# Nanofluidic chips for cryo-EM structure determination from picoliter sample volumes

Stefan T Huber[1], Edin Sarajlic[2], Roeland Huijink[2], Felix Weis[3], Wiel H Evers[1], Arjen J Jakobi[1]*

[1]Department of Bionanoscience, Kavli Institute of Nanoscience, Delft University of Technology, Delft, Netherlands; [2]SmartTip B.V., Enschede, Netherlands; [3]Structural and Computational Biology Unit, European Molecular Biology Laboratory (EMBL), Heidelberg, Germany

**Abstract** Cryogenic electron microscopy has become an essential tool for structure determination of biological macromolecules. In practice, the difficulty to reliably prepare samples with uniform ice thickness still represents a barrier for routine high-resolution imaging and limits the current throughput of the technique. We show that a nanofluidic sample support with well-defined geometry can be used to prepare cryo-EM specimens with reproducible ice thickness from picoliter sample volumes. The sample solution is contained in electron-transparent nanochannels that provide uniform thickness gradients without further optimisation and eliminate the potentially destructive air-water interface. We demonstrate the possibility to perform high-resolution structure determination with three standard protein specimens. Nanofabricated sample supports bear potential to automate the cryo-EM workflow, and to explore new frontiers for cryo-EM applications such as time-resolved imaging and high-throughput screening.

*For correspondence:
a.jakobi@tudelft.nl

## Editor's evaluation

In their pioneering work, the authors describe a prototype of a new microfluidic device that addresses several of the common sample preparation problems in single-particle cryo-EM. By providing small cavities made of thin silicon nitride membranes, picoliters of sample can be frozen with defined ice thickness while avoiding sample damage at the air-water interface. The new approach requires further development to become mainstream but has the potential to become a powerful alternative to the conventional workflow of single-particle cryo-EM, enabling full automation and making sample preparation highly reproducible.

## Introduction

Sample preparation of frozen-hydrated biological macromolecules continues to present a major challenge for routine structure determination by three-dimensional cryogenic transmission electron microscopy (cryo-EM). To obtain a three-dimensional structure, a thin film of an aqueous solution containing the protein of interest must be cooled so rapidly that water molecules do not crystallise but form amorphous ice. Within this layer, ideally not much thicker than the particle itself, the particles must be partitioned in random orientations. The prevailing method for making thin specimens suitable for cryo-EM has not changed substantially since its introduction several decades ago: a filter paper is pressed against one or both sides of a fenestrated support film spanning a metallic mesh grid across which a droplet of the biological sample is suspended, and then rapidly plunged into a

liquid cryogen (*Adrian et al., 1984*). A recognised shortcoming of this method is that a reproducibly thin and consistently uniform ice layer can often not be achieved (*Carragher et al., 2019*; *Armstrong et al., 2020*). Frequently, time-consuming parameter optimisation is required to find conditions suitable for high-resolution imaging, and these conditions are typically different for each sample. Even if such conditions are found, additional problems encountered in practice include preferential orientation of particles (*Tan et al., 2017*; *Noble et al., 2018*), preferential interaction of particles with the support film (*Meyerson et al., 2014*) and partial denaturation of particles due to exposure to the large air-water interface (*Donaldson et al., 1980*; *Glaeser and Han, 2017*; *Glaeser, 2018*; *Noble et al., 2018*; *D'Imprima et al., 2019*).

The inefficiency of current cryo-EM sample preparation approaches to yield reproducible results has led major efforts to explore different ways to make uniformly thin films, including the development of self-wicking grids (*Wei et al., 2018*), spraying small-volume droplets onto holey support films (*Feng et al., 2017*; *Rubinstein et al., 2019*; *Kontziampasis et al., 2019*) and direct thin film dispensing onto cryo-EM grids (*Jain et al., 2012*; *Arnold et al., 2017*; *Ravelli et al., 2020*). These developments have increased the speed of sample preparation and reduced the required sample volume by orders of magnitude. Perhaps surprisingly though, the core design of fenestrated EM grid supports has remained largely unchanged. Despite the accelerated sequence from sample deposition to vitrification, the exposure of particles to the air-water interface inherent to grid-based sample supports continues to pose a major obstacle for robust sample preparation workflows (*Klebl et al., 2020*; *Glaeser, 2021*).

The detrimental effects of an air-water interface could be avoided entirely if the sample were encapsulated within an enclosure made of a substrate sufficiently electron-transparent for high-contrast imaging of weak phase objects. Liquid-phase transmission electron microscopy (LP-EM) has been widely applied to image liquids inside the vacuum of an electron microscope by enclosing them in liquid cells (*Peckys et al., 2009*; *Nishiyama et al., 2010*). Microfabrication of LP-EM sample cells has drawn on advances in semiconductor-manufacturing processes, in particular the use of free-standing silicon nitride (SiN) membranes (*Ciarlo, 2002*; *Ring et al., 2011*; *Gorelick et al., 2019*). This approach has also been successfully employed for room temperature LP-EM of biological macromolecular specimens (*Mirsaidov et al., 2012*; *Cameron Varano et al., 2015*). To render liquid cells mechanically robust against elastic deformation caused by the pressure differential between the fluid-filled cell and the vacuum in the microscope column, LP-EM cells contain relatively thick (~30–50 nm) SiN membranes, whose additional scattering contributes structural noise that limits the attainable resolution.

In vitrified samples, the solidified water behaves as an ultra-viscous fluid (*Angell, 2002*). In this case, elastic deflection of fluidic cells under a pressure difference is negligible, which offers opportunities for employing much thinner cells and membranes. Inspired by this prospect, we set out to design a nanofluidic sample cell for cryo-EM imaging. By miniaturising a sample-containing volume into a microelectromechanical system (MEMS) with ultrathin SiN membranes, we fabricated a cryo-EM sample support that allows cryo-EM imaging from picoliter sample volumes. We demonstrate the performance of this device by high-resolution 3D structure determination of three protein model systems, apoferritin, T20S proteasome, and tobacco mosaic virus.

## Results

### A nanofluidic sample support for cryo-EM imaging

To address common issues in cryo-EM grid preparation such as poor reproducibility of ice thickness, the destructive effect of the air-water interface and to broaden the scope of cryo-EM applications with the unique design flexibility of MEMS technology, we developed a novel cryo-EM sample support using nanofabrication techniques. We used a sacrificial polysilicon process for fabrication of a device featuring nanofluidic channels for controlled cryo-EM sample preparation and imaging (*Figure 1*, *Figure 1—figure supplement 1*). The nanochannels are enclosed by ultrathin membranes made from silicon-rich nitride ($SiN_x$) that allow for transmission of the electron beam when a thin film containing the sample is encapsulated in the space between them. The core design of this device, which we refer to as cryoChip, consists of a 2 × 2 mm silicon base with 100 μm thickness (*Figure 1a*). A hollow microcantilever extends over the side of the chip base for sample application via capillary action.

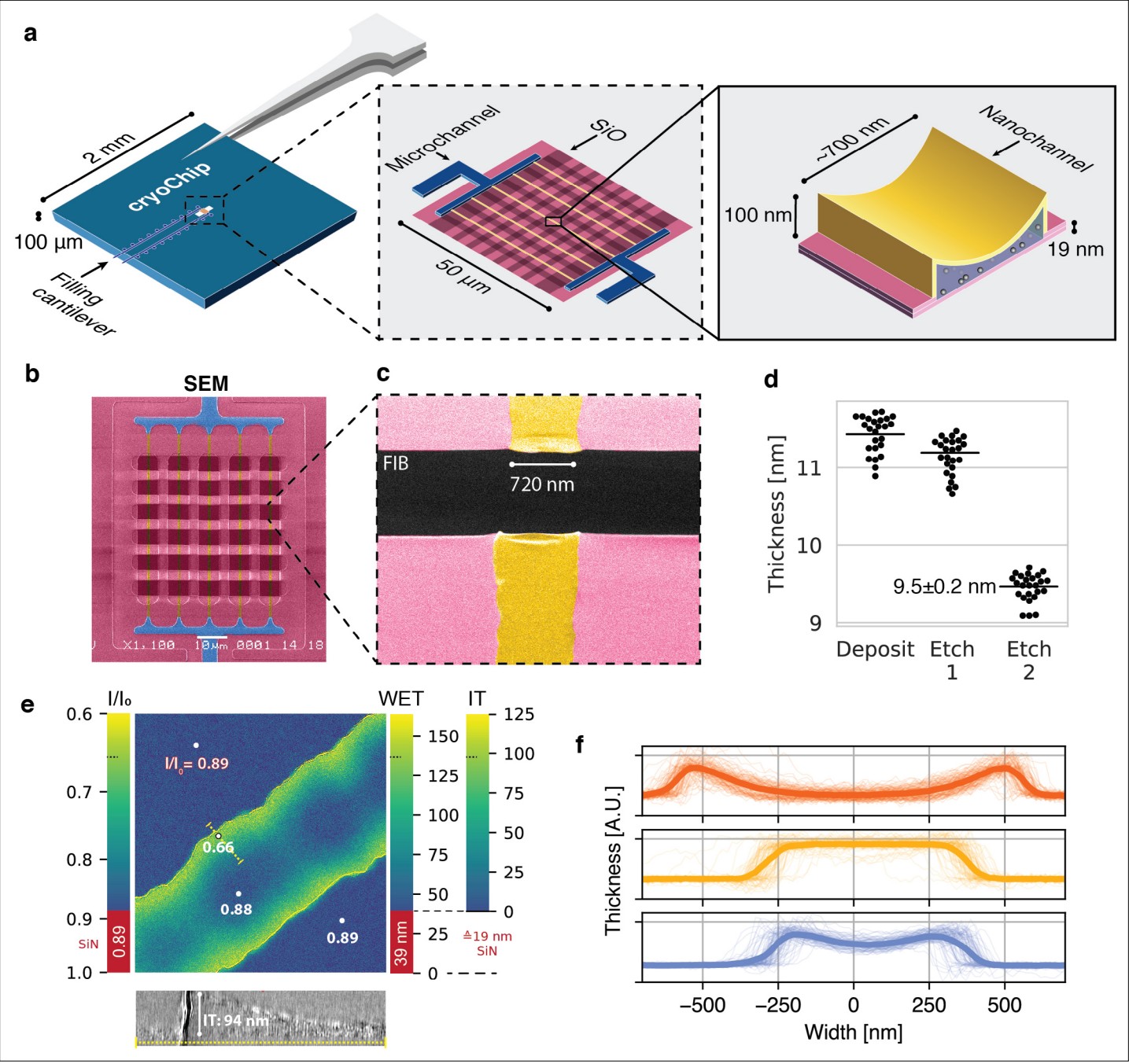

**Figure 1.** A nanofluidic sample support for cryo-EM. (**a**) Schematic representation of the MEMS-based cryoChip. The 2 × 2 mm chip base features two microcantilevers for sample filling and air escape. The cantilevers are connected to microchannels leading to a free-standing observation membrane located within an aperture at the chip center (middle inset). A multifurcation connects the microchannels (blue) to an array of five parallel nanochannels (yellow) of approximately 700 nm width and 100 nm height (rightmost inset). The top membrane is smoothly sagging, while the bottom membrane is flat. (**b**) False-colour scanning electron microscopy (SEM) image of the observation membrane. (**c**) False-colour SEM image of a single nanochannel with FIB-milled rectangular opening across the observation membrane (**d**) Ellipsometric determination of $SiN_x$ membrane thickness after deposition and etching steps. (**e**) Relative thickness map of an ice-filled nanochannel from zero-loss imaging. WET = water-equivalent thickness, IT = ice thickness. The thickness profile correlates with a 2D slice of a tomogram acquired of the same area (bottom). (**f**) Thickness cross-sections of nanochannels from three chips of the same wafer. Thin lines are individual nanochannel cross-sections measured across a whole chip; thick lines represent their mean.

The online version of this article includes the following source data and figure supplement(s) for figure 1:

**Figure supplement 1.** cryoChip fabrication.

*Figure 1 continued on next page*

*Figure 1 continued*

**Figure supplement 2.** Specimen mounting.

**Source data 1.** Raw ellipsometry data.

An aperture in the center of the silicon base exposes a 50 × 50 µm free-standing $SiN_x$ membrane containing parallel rows of nanochannels that are connected to the central supply and exit channels via multifurcations. Transversal silicon oxide layers provide mechanical stability and generate a regular grid of observation windows that are 6 × 4 µm in size (*Figure 1b*). Each observation window exposes a segment of the nanochannel for TEM imaging (*Figure 1c*). CryoChips differ in several fundamental aspects from grid-based sample supports: Instead of wicking a suspended droplet to a liquid film thin enough for transmission imaging, sample thickness in cryoChips is controlled by the geometry of electron-transparent nanochannels itself. The closed channel architecture avoids the air-water interface, which is replaced by a solid-liquid interface between the $SiN_x$ membrane and the sample solution. With minimal adaptions, cryoChips are fully compatible with the most frequently used cryo-EM sample carrier cartridges (*Figure 1—figure supplement 2*).

## Geometric control of ice thickness

Control over ice thickness is a critical step for cryo-EM sample preparation. In standard blotting-based sample preparation, parameters such as blot force and time, absorbent properties of the filter paper, as well as relative humidity and temperature of the sample environment need to be tightly controlled. Despite the availability of robotic vitrification systems, achieving good reproducibility of these parameters has remained difficult and substantial across-grid and grid-to-grid variation of ice quality are common (*Carragher et al., 2019*). MEMS-based nanofabrication provides a means to control sample thickness directly through channel geometry, obviating the need for a particular sample environment and optimisation of blotting conditions. To quantify the reproducibility of ice thickness in cryoChip nanochannels, we first determined the average thickness of the silicon nitride layer by replicating the $SiN_x$ deposition and etching protocol employed for device fabrication on a blank wafer area. Using ellipsometry we measured a film thickness of 9.5 ± 0.2 nm for the final $SiN_x$ layer (*Figure 1d*). We then filled the chip and used zero-loss imaging to determine the thickness of ice-filled nanochannels from the ratio of the integrated intensity of the zero-loss peak $I_0$ relative to the integral of the whole spectrum $I$ (*Figure 1e*). From the zero-loss log ratio of a $SiN_x$ area adjacent to the nanochannel (0.89) and the geometric film thickness determined by ellipsometry (19 nm, corresponding to the combined thickness of top and bottom membrane), we calculated the mean free path for inelastic scattering (IMFP) at 300 keV in $SiN_x$ as 163 nm, from which we estimate a combined water-equivalent thickness of 39 nm for the two $SiN_x$ membranes forming the nanochannel assuming an IMFP of 320 nm for vitreous ice (*Yesibolati et al., 2020*). We next mapped the apparent and absolute ice thickness at each image position. The channels display a concave profile in cross-section, leading to a gradient with a maximum thickness of ~100 nm at the channel wall and 0 nm at the channel center. These estimates are in agreement with those obtained from a tomographic reconstruction from the same image area (*Figure 1e*). We evaluated the consistency of nanochannel geometry both across individual cryoChips and between different chips. The precision of the sacrificial layer deposition in the fabrication process permits tight control over the maximum channel height. The width of the nanochannels is consistent across individual chips, but can range from 650 nm to 1000 nm between different chips which we attribute to process variability and/or the chip position on the wafer during fabrication (*Figure 1f*).

## Rapid, loss-less preparation of cryo-EM samples via microcantilevers

To facilitate filling of the device from nanoliter sample droplets, we designed a hollow microchannel cantilever that allows sample application via spontaneous capillary flow without requirement for external appliances. The hollow microchannel cantilever extends only 100 µm over the chip base. To allow efficient targeting of the cantilever, we built a loading station consisting of a microvolume positive displacement syringe mounted on a linear XYZ precision stage, as well as top and side-view cameras to monitor the approach of the nozzle and filling process (*Figure 2a*). The syringe nozzle is translated into close vicinity of the cantilever tip and a nanoliter droplet is formed. Manual handling of the syringe and the geometry of the nozzle currently limit the minimum droplet volume to about

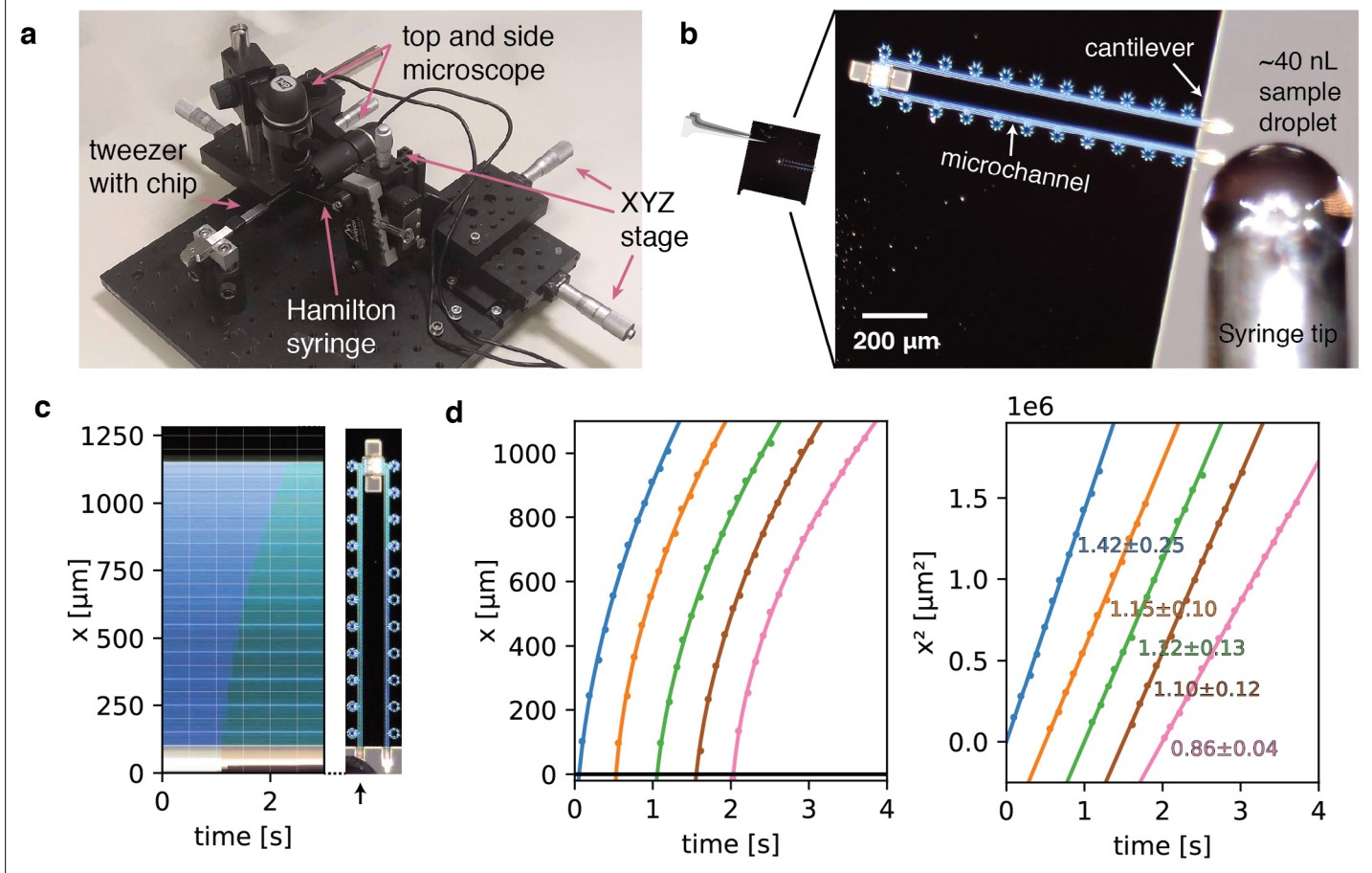

**Figure 2.** Sample preparation with cryoChips. (**a**) Manual filling station with side view and top view microscope, microvolume glass syringe, xyz precision stages for translation of the syringe and a mount for tweezers holding a cryoChip. (**b**) Sample loading of a cryoChip seen with the top view microscope. One of the cantilevers is approached with a ~40 nL sample droplet. (**c**) Kymograph of the filling process showing the progression of the sample meniscus over time. The arrow indicates the cantilever and the supply channel. (**d**) Filling kinetics of cryoChips. The meniscus position and associated time value were recorded at 10 fps with the top-view microscope, revealing a decreasing filling rate with progression of the fluid front (left panel). Each graph represents data from one filling experiment. Graphs from individual experiments are spaced apart by 0.5 s for better visualisation. The squared meniscus position (right panel) displays a linear relationship with elapsed filling time in agreement with expectation from Washburn kinetics. Solid lines are non-linear fits to the data. Slopes ($10^6$ $\mu m^2$/s) and their standard deviations from bootstrapping statistics are shown.

The online version of this article includes the following video and figure supplement(s) for figure 2:

**Figure supplement 1.** Filling kymographs.

**Figure 2—video 1.** Demonstration of cryoChip filling.
https://elifesciences.org/articles/72629/figures#fig2video1

**Figure 2—video 2.** Raw movie for kymograph (a).
https://elifesciences.org/articles/72629/figures#fig2video2

**Figure 2—video 3.** Raw movie for kymograph (b).
https://elifesciences.org/articles/72629/figures#fig2video3

**Figure 2—video 4.** Raw movie for kymograph (c).
https://elifesciences.org/articles/72629/figures#fig2video4

**Figure 2—video 5.** Raw movie for kymograph (d).
https://elifesciences.org/articles/72629/figures#fig2video5

**Figure 2—video 6.** Raw movie for kymograph (e).
https://elifesciences.org/articles/72629/figures#fig2video6

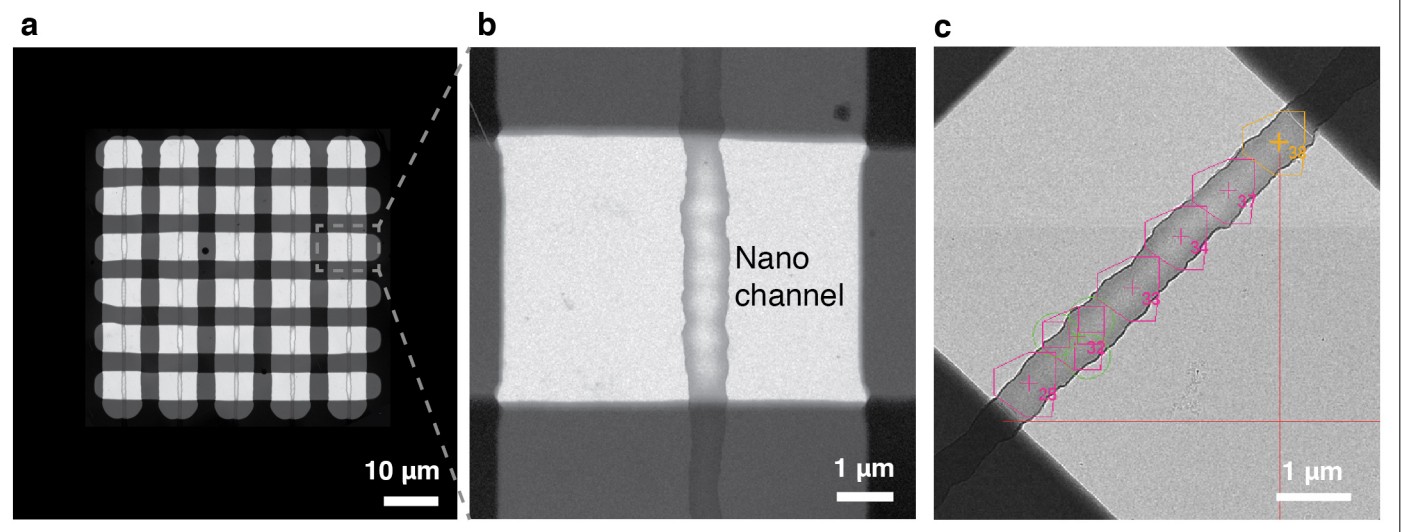

**Figure 3.** Automated data acquisition with cryoChips. (**a**) TEM image of the observation membrane with five nanochannels. The checkerboard structure is a stiffening layer made from silicon oxide that divides the nanochannels in observation windows. (**b**) Close-up of inset in (**a**). (**c**) Medium magnification image of an observation square with an overlay of the SerialEM multishot acquisition scheme containing six acquisition points along the nanochannel. Each acquisition point is centered on a polygon outlining the limits of three imaging positions achieved by beam-image shift, leading to a total of 18 images per observation square.

10–40 nL (*Figure 2b*). The cantilever is then dipped into the nanodroplet. Upon contact with the cantilever tip, a small sample volume is drawn into the cantilever by capillary forces and subsequently distributed across the central nanochannels via the multifurcation connected to the entry micro-channel (*Figure 2—video 2*). From the geometry of the overall channel design, we estimate the total filling volume to be ~4 pL (see Materials and methods), which is about six orders of magnitude less compared to typical sample volumes of conventional blotting-based methods. Residual sample of the filling droplet can be readily recovered.

To analyse fluid kinetics in the microchannels, we filled a series of cryoChips via the sample applica-tion cantilever to generate kymographs and determined the capillary flow rate by monitoring the posi-tion of the filling meniscus over time. We observed the movement of the liquid front to be proportional to the square root of time, which is in accordance with classical Washburn kinetics for capillary flow (*Washburn, 1921*). Filling is rapid and complete filling of the entry channel proceeds at an average of ~1 mm/s (*Figure 2c*). The filling speed is fast at the beginning and slows down with progression of the liquid front, as predicted by the Washburn equation. We found similar behaviour across the chips that were analysed, suggesting that we can achieve high channel-to-channel reproducibility during fabrication (*Figure 2d*, *Figure 2—figure supplement 1*). The entire filling process can be executed in a lab environment without humidity control and there is no further requirement for parameter optimisation other than sample concentration. After sample filling, the chip can be transferred to any manual or robotic vitrification device for flash cooling in liquid cryogen. In our experiments, the total time from filling to vitrification was approximately 10 s. This process could be readily accelerated by integrating the filling mechanism directly on a vitrification system.

## Automated cryo-EM data collection with cryoChips

The checkerboard pattern from the silicon oxide layers on the observation membrane structures the nanochannel array into a regular grid of 30 observation squares, each comprising a segment of an individual nanochannel and flanking areas of the SiN$_x$ membrane (*Figure 3a and b*). Due to the different layout of the cryoChip, standardised data acquisition protocols established for holey support film substrates cannot be applied without modification. To develop an automated data acquisition strategy for cryoChips, we first acquired a grid map of the entire observation membrane using Seri-alEM (*Mastronarde, 2005*). For each observation square we then selected six equally spaced acqui-sition points positioned linearly along the midline of the nanochannel (*Figure 3c*). We used virtual maps (*Schorb et al., 2019*) generated from medium magnification images to accurately realign to

the acquisition points during high-magnification acquisition. For each of the acquisition points, three images were acquired radially around the midpoint using beam-image shift, resulting in up to 540 images per cryoChip.

## 3D structure determination from cryoChip samples

To validate the suitability of cryoChips for 3D structure determination, we collected datasets for three test specimen (*P. furiosus* ferritin (ApoFtn, 480 kDa, O), *T. acidophilum* 20 S proteasome (T20S, 700 kDa, D7) and tobacco mosaic virus (TMV, 37 MDa, helical)) that we selected for differences in size, shape and symmetry. The low magnification maps revealed that most observation squares were suitable for data collection. We occasionally observed partially broken or cracked membranes. Observation squares in such areas were omitted for data collection. Micrographs for all three specimens showed good image contrast (*Figure 4a*). To examine the compatibility of cryoChips with the condition to form vitreous ice, we evaluated the fraction of images containing crystalline ice by manual inspection for the presence of ice rings or crystalline reflections in the computed power spectra from all images. This led to a fraction of 86% (815/945), 75% (91/121), and 52% (43/82) of images that we classified as containing vitreous ice for ApoFtn, T20S and TMV respectively (*Figure 4e*, *Figure 4—figure supplement 1*). This is consistent with convective heat-transfer simulations of a simplified cryoChip model, suggesting that cooling below the glass-transition temperature occurs at timescales (< $10^{-5}$ seconds) compatible with vitrification (*Figure 4—figure supplement 2*). It is possible that the fraction of vitreous images is underestimated since we did not discriminate between ice features originating from surface contamination and those originating from crystalline ice in the nanochannels. To investigate whether non-vitreous images are a result of inefficient vitrification or phase transition to cubic ice during exposure, we analysed a subset of images classified as icy. From a maximum intensity projection of all power spectra of all images, we find evidence for hexagonal ice indicating incomplete vitrification (*Figure 4—figure supplement 3*). We did not observe evidence for a correlation of the probability for forming crystalline ice and the location on the observation membrane.

The SiN$_x$ membrane forming the walls of the nanochannels leads to additional background noise from an increase in the fraction of electrons that are scattered inelastically. This effect is equivalent to an apparent increase in ice thickness, which even for particles in the thinnest parts of the nanochannel may lead to an (apparent) ice thickness significantly greater than the size of the particle itself. This is undesirable because it leads to a corresponding loss of useful signal in the image, thereby reducing image contrast. To test whether this additional scattering limits our ability to extract high-resolution signal for particle alignment, we used automatic particle picking in cryoSPARC 3.1 (*Punjani et al., 2017*) to extract particles (segments for TMV) and performed 2D classification. For all three specimens, we observe high-contrast 2D class averages with visible features indicative of secondary structure (*Figure 4b*, *Figure 4—figure supplement 4*, *Figure 4—figure supplement 5*, *Figure 4—figure supplement 6*), suggesting that images contain sufficient contrast for high-resolution structure determination. To probe the resolution limit of data collected using the cryoChip, we merged data from three chips filled with ApoFtn acquired on a Titan Krios equipped with a post-GIF K2 camera and processed it in a standard single-particle analysis workflow. We used 2D and 3D classification to select a final set of 21,238 particles (O symmetry) from 815 micrographs and after 3D refinement obtained a reconstruction at an overall resolution of 3.0 Å displaying visible side chain densities (*Figure 4c/d*). The reference model for 3D alignment was generated from the data itself using the stochastic gradient descent method (*Figure 4—figure supplement 4*), suggesting that the quality of data collected from cryoChips is sufficient to allow initial model generation even for samples considered difficult for reference model generation. To demonstrate the versatility of cryoChips for structure determination from a variety of biological specimens, we also processed the datasets for TMV and T20S proteasome (*Figure 4—figure supplement 5*, *Figure 4—figure supplement 6*). These datasets were collected on a JEOL JEM3200-FSC with a K2 detector and yielded 3D reconstructions with overall resolutions of 3.7 Å and 5.4 Å obtained from 14,238 segments (20 asu/segment) and 5750 particles (D7), respectively (*Figure 4f*, *Supplementary file 1*). The quality of a dataset is often quantified from a plot of the inverse-squared resolution as a function of the number of particles used in the 3D reconstruction. The slope of this function is related to a B-factor that describes the width of a Gaussian dampening function in reciprocal space used to model the loss of high-resolution information in particle images owing to a combined effect of performance-reducing factors related to the sample, data collection and

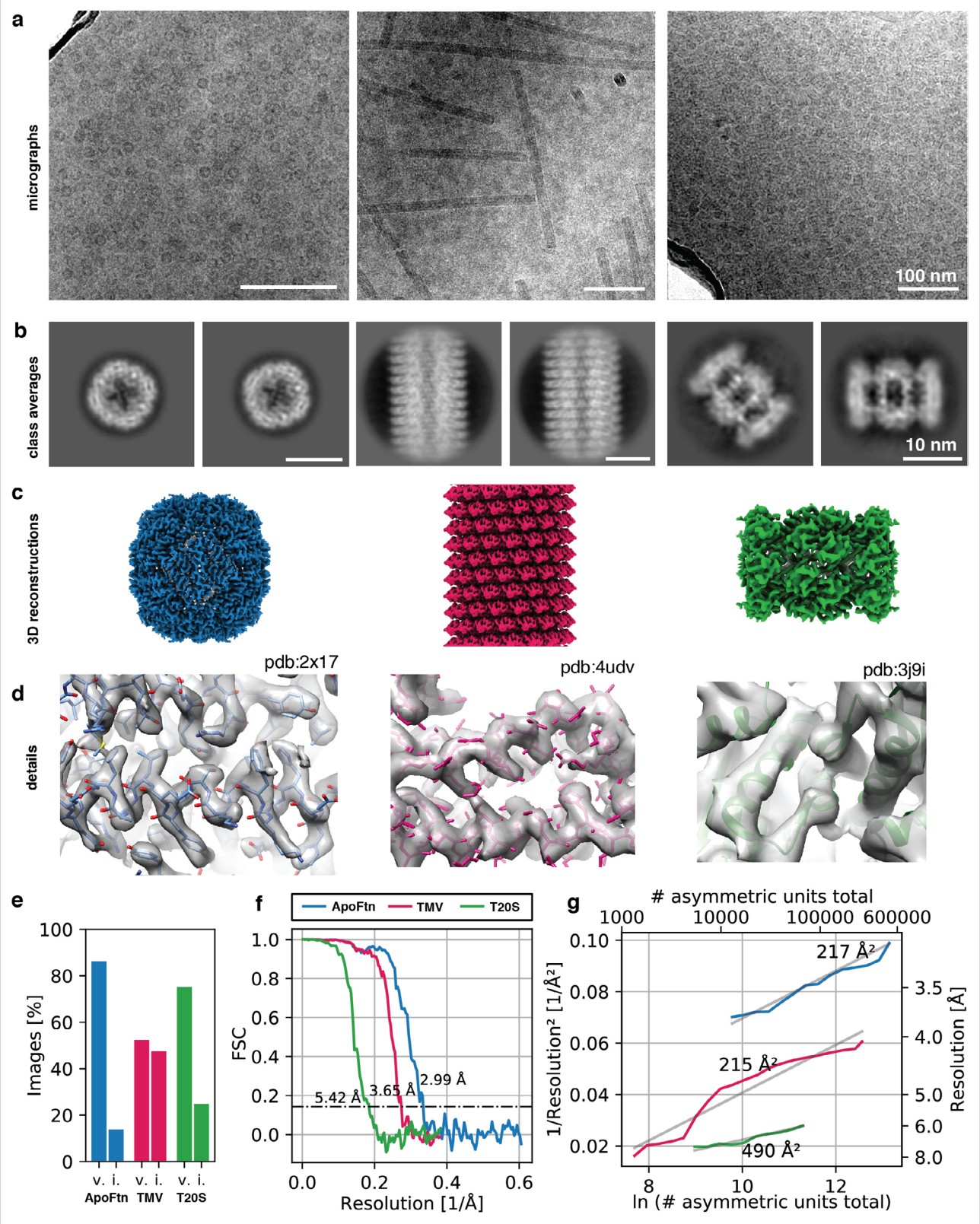

**Figure 4.** 3D reconstruction of ApoFtn, T20S proteasome and TMV. (**a**) Raw micrographs, (**b**) 2D class averages, (**c**) 3D cryo-EM maps and (**d**) close-up densities displaying discernible details for the three test specimens. Atomic models were overlaid as visual aid: ApoFtn pdb:2×17 (*Kasyutich et al., 2010*), TMV pdb:4udv (*Fromm et al., 2015*), T20S pdb:3j9i (*Li et al., 2013*) (**e**) Percentage of images in the dataset classified as vitreous (**v**) and icy (**i**). (**f**) Fourier Shell Correlation (FSC) curves with 0.143 cutoff. (**g**) ResLog plots for reconstructions. B-factors were estimated as two over the fitted slope.

*Figure 4 continued on next page*

*Figure 4 continued*

Colour coding is equivalent for panels (**c–g**). Micrographs in (**a**) were taken at a defocus of –1.2 µm for ApoFtn, –1.8 µm for TMV and –2.8 µm for T20S. Scale bars are equivalent for images in (**a**) and class averages in (**b**).

The online version of this article includes the following source data and figure supplement(s) for figure 4:

**Source data 1.** Raw data for vitrification statistics.

**Source data 2.** Raw data Fourier Shell Correlation (FSC) curve for ApoFtn dataset.

**Source data 3.** Raw data Fourier Shell Correlation (FSC) curve for T20S dataset.

**Source data 4.** Raw data Fourier Shell Correlation (FSC) curve for TMV dataset.

**Source data 5.** Raw data B-factor plot for T20S dataset.

**Source data 6.** Raw data B-factor plot for TMV dataset.

**Source data 7.** Raw data for COMSOL simulation.

**Figure supplement 1.** Vitrification efficiency.

**Figure supplement 2.** COMSOL simulations.

**Figure supplement 3.** Ice formation.

**Figure supplement 4.** ApoFtn processing.

**Figure supplement 5.** TMV processing.

**Figure supplement 6.** T20S processing.

**Figure supplement 7.** B-factor analysis.

instrumentation, and data processing (*Rosenthal and Henderson, 2003*). We quantified the B-factors for each of the three datasets (*Figure 4g*). For comparison, we also acquired datasets for ApoFtn and T20S on conventional holey carbon supports with the same acquisition parameters, yielding reconstructions at an overall resolution of 2.4 and 3.2 Å (*Figure 4—figure supplement 7*, *Supplementary file 2*). This leads to B-factors of 217 Å$^2$ vs. 178 Å$^2$ for ApoFtn and 490 Å$^2$ vs. 328 Å$^2$ for T20S determined for cryoChip and holey carbon datasets, respectively. We also note a consistent upward shift of the offset for holey carbon B-factor plots compared to those from cryoChips, consistent with the differences in overall signal-to-noise ratio caused by additional scattering from the SiN$_x$ membrane.

## Beam-induced specimen motion is comparable to standard cryo-EM supports

Beam-induced motion is a major factor limiting the achievable resolution of 3D reconstructions. The highest rate of change in particle position is typically largest at cumulative exposures below ~4e⁻/Å$^2$ (*Vinothkumar and Henderson, 2016*), coinciding with the frames where the sample is yet least affected by radiation damage and high-resolution signal is best preserved. The fast read-out rates of direct electron detection technology have made it possible to acquire cryo-EM data as dose-fractionated movies where the total exposure is subdivided into a series of individual frames. To reduce information loss, alignment of these frames can be used to determine the overall inter-frame movement of the specimen in the micrograph during the exposure and then correct for this motion before summation of the frame stack. To investigate beam-induced sample movement in cryoChips, we computed and analysed the alignment trajectories for the ApoFtn dataset (*Figure 5a*). The calculated traces show that movement is largely unidirectional, but clusters into several distinct directions of motion. To further investigate the trajectories, we separated them either by chip or by the beam-image shift condition corresponding to the shot position in the multishot acquisition scheme for each acquisition point (*Figure 5b*, *Figure 5—figure supplement 1*). For each of the chips analysed, we observe three clusters of directional motion, each associated with one of the three shot positions. While the precise origin for this clustering needs further investigation, we suggest that different thickness and geometry of the confined ice layer in the subregions for each of the multishot positions may lead to preferential directionality of specimen movement, possibly combined with microlensing effects resulting from adjacent exposures that induce the build-up of semi-static charge on the non-conductive SiN$_x$ membrane.

To compare average specimen movement in cryoChips with other commonly used specimen supports we also computed the alignment trajectories for four datasets in the EMPIAR database

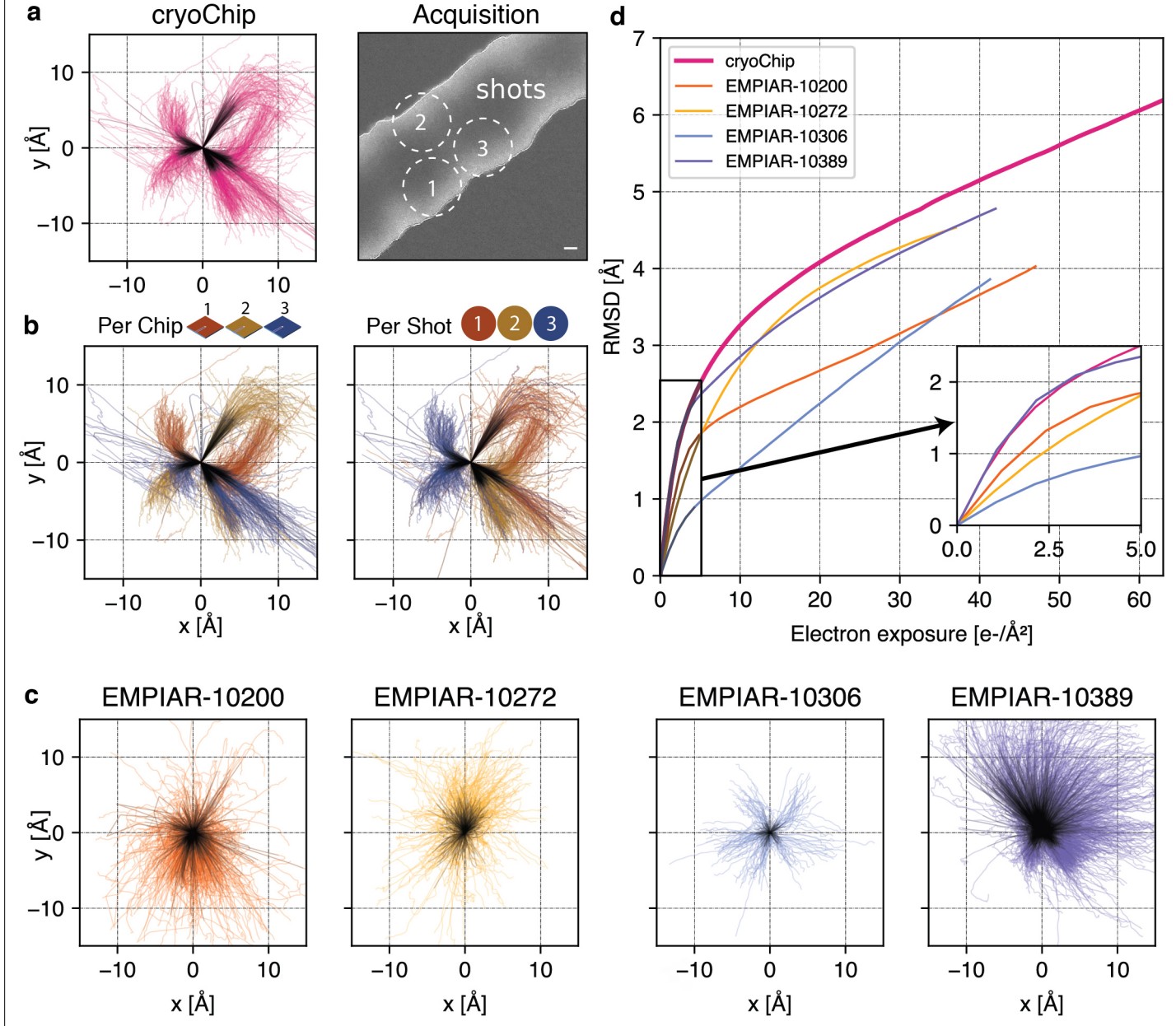

**Figure 5.** Comparison of beam-induced motion for cryoChip and reference datasets from conventional sample preparation. (**a**) Whole-frame motion trajectories of dose-fractionated movies during exposure with 63 e⁻/Å² from three different cryoChips. (**b**) Trajectories coloured by chip or by imaging position in a repeating three-shot pattern. (**c**) Whole-frame motion trajectories of reference datasets. (**d**) Root-mean-square deviation (RMSD) of trajectories over electron exposure between the cryoChip dataset and reference datasets. The inset shows the first part of the exposure most critical for reconstructing high-resolution details. Scale bar in (**a**) is 100 nm. First five e⁻/Å² in trajectories are coloured in black.

The online version of this article includes the following source data and figure supplement(s) for figure 5:

**Source data 1.** Raw motion track data for cryoChip 1,2, and 3.

**Source data 2.** Raw motion track data for EMPIAR 10200, 10272, 10306, 10,389.

**Source data 3.** Raw data for cumulated motion plots.

**Figure supplement 1.** Motion analysis.

**Figure supplement 1—source data 1.** Raw motion track data for *Figure 5—figure supplement 1*.

(*Iudin et al., 2016*) and ApoFtn on holey carbon (*Figure 5c*, *Figure 4—figure supplement 7b,c*). These datasets have been acquired using different specimen supports but resulted in high-resolution 3D reconstructions (*Supplementary file 4*). We then computed the accumulated movement as the root mean squared deviation (RMSD) of all trajectories from the ApoFtn cryoChip dataset and the EMPIAR datasets and evaluated them over an equivalent exposure range (*Figure 5d*). The datasets show small differences in accumulated movement over the whole exposure, ranging from 3.8 (QF-R2/2) to 5.1 Å (cryoChip) RMSD over 0–40 e⁻/Å². These movements are in the order of 0.1 pixel per frame and are therefore amenable to motion correction. These overall motions represent a combination of mechanical (stage) drift and beam-induced motion. Limiting motion in early frames is essential to maximise recovery of high-resolution detail. EMPIAR-10306 has a remarkably low initial movement of 0.7 Å, while both our dataset and EMPIAR-10389 display a accumulated motion of 1.8 Å. EMPIAR-10389 led to a 2.0 Å reconstruction of *Y. enterocolitica* urease, indicating that initial specimen movement of that order is not limiting for achieving resolution beyond 2.0 Å.

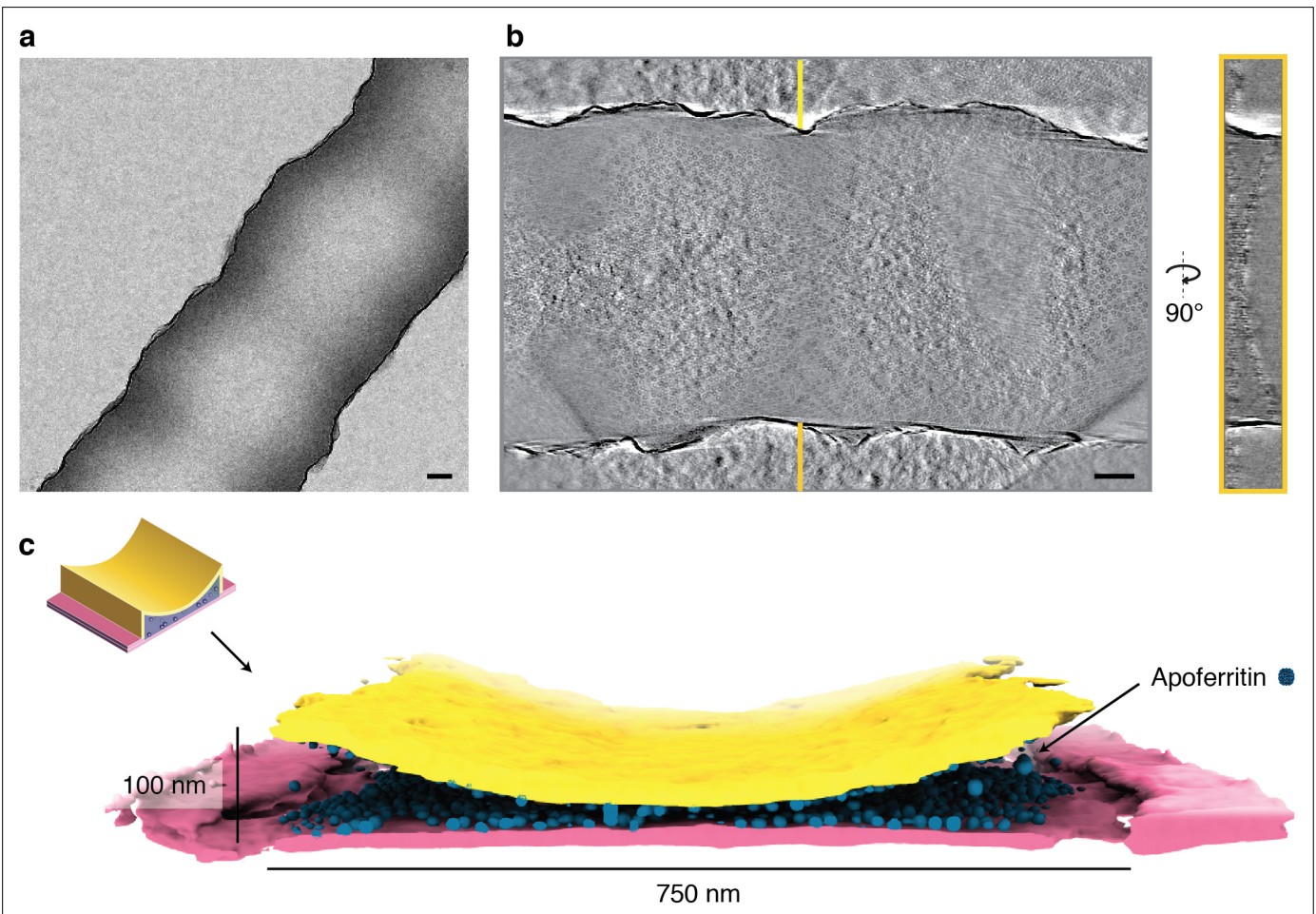

**Figure 6.** Tomographic reconstruction of a cryoChip nanochannel filled with ApoFtn. (**a**) Overview image of tomogram area of a nanochannel segment filled with ApoFtn. (**b**) Slice through the reconstructed tomogram and yz cross-section across the nanochannel indicated by lines. (**c**) Segmented reconstruction showing the nanochannel membrane (bottom: pink, top: yellow) and ApoFtn in the channel lumen (blue). The thickness of the membrane is affected by the missing wedge and not to scale. All scale bars: 100 nm.

The online version of this article includes the following video and figure supplement(s) for figure 6:

**Figure supplement 1.** TMV and T20S cryoChip tomograms.

**Figure supplement 2.** SiN Fourier spectra.

**Figure supplement 3.** Surface electrostatic potential maps.

**Figure 6—video 1.** Zoom-in video of cryoChip.

https://elifesciences.org/articles/72629/figures#fig6video1

## Preferential partitioning to the solid-liquid interface effectively concentrates protein particles

Interaction with the air-water interface may cause perturbation of molecular structure in thin film-based cryo-EM sample preparations. The closed sample environment of the cryoChip avoids the problems of an air-water interface, but the large surface-to-volume ratio of the nanochannels forms an extensive solid-liquid interface between the $SiN_x$ surface and the particle solution. We performed tomographic reconstructions of protein-loaded cryoChips to investigate particle distributions inside the nanochannels. The tomographic reconstructions confirm the overall geometry of the channels with an estimated height in cross-section ranging between 0 and 100 nm and a channel width of ~750 nm (*Figure 6a/b*). In all cases, we find the majority of particles close to or associated with the $SiN_x$ membrane (*Figure 6c*, *Figure 6—figure supplement 1*, *Figure 6—video 1*). Analysis of the orientation distribution showed increased abundance of views along the fourfold rotation axis for ApoFtn, indicating preferential alignment of particles with the $SiN_x$ membrane (*Figure 4—figure supplement 4*). Similarly, we find preferential abundance of side views for T20S particles (*Figure 4—figure supplement 6*). We did not observe convincing correlation of preferred orientations with the molecular electrostatic potential (*Figure 6—figure supplement 3*). We note that the current analysis is limited to particles with internal symmetry for which preferred orientations are less problematic. Further investigation will be required to conclude whether systematic effects for preferred protein adsorption to the solid-liquid interface exist and need to be addressed. Micrographs from datasets of all three specimen display particle densities exceeding those expected from conventional cryo-EM grids prepared with the same specimens and sample concentrations. To quantify this concentration effect, we estimated the particle concentration in the nanochannels by segmenting and counting ApoFtn particles, and measuring the channel volume from the reconstructed tomogram (*Figure 6—figure supplement 1*). We find 1483 ApoFtn particles within a $9.4 \times 10^6$ $nm^3$ volume, equivalent to a concentration of 262 µM or 126 mg/mL assuming 480 kDa molecular mass. This is a 37-fold increase in concentration compared to the initial sample concentration of 3.4 mg/mL. Interaction with the solid-liquid interface at the $SiN_x$ channel walls thus strongly affects the effective sample concentration in the nanochannels. Similar concentration effects have also been reported for traditional sample preparation methods, such as repeated sample application (*Snijder et al., 2017*) and in the presence of very thin ice for which densely packed protein monolayers have been observed (*Yip et al., 2020*).

## Discussion

The process of sample preparation is widely perceived as a major bottleneck for realising the full potential of cryo-EM. All existing approaches to cryo-EM sample preparation rely on the ill-controlled formation of thin ice layers across fenestrated support films, which results in poor reproducibility and may detrimentally affect particle distribution, orientation and integrity (*D'Imprima et al., 2019*; *Noble et al., 2018*; *Glaeser and Han, 2017*; *Glaeser, 2018*). Recent improvements to the support grids and to how the sample is applied have begun to address these issues (*D'Imprima et al., 2019*; *Naydenova et al., 2019*; *Tan et al., 2017*; *Ravelli et al., 2020*; *Jain et al., 2012*; *Arnold et al., 2017*; *Drulyte et al., 2018*). Despite these advances important issues remain, most prominently the hydrophobic environment of the air-water interface (*Klebl et al., 2020*; *Glaeser, 2021*). Here, we have shown that MEMS-based nanofluidic chips in which the sample is enclosed and protected from the air-water interface within electron-transparent nanochannels provide a viable alternative for cryo-EM sample preparation that mitigates some of the prevalent issues observed for grid-based sample supports. We demonstrated that biological specimens in cryoChips can be successfully vitrified and that the thickness uniformity of the ice layer can be reproducibly controlled by channel geometry. We found the width and height of the nanochannels to be highly consistent across individual cryoChips, providing uniform imaging conditions for all parts of a chip.

The most widely applied methods for cryo-EM sample preparation currently require ~3 µL of sample volume per grid. Assuming a final liquid film of 20–100 nm thickness spanning a 3.05 mm EM support grid, the required sample volume for imaging is only 1.5–7.0 fL, which is nine orders of magnitude less than the initial sample volume. Several recent developments are trying to reduce this gap: The Chameleon/Spotiton instrument eliminates sample waste from the blotting step by applying nanoliter sample volumes onto self-wicking grids via a piezo-electric droplet dispenser (*Jain et al., 2012*; *Wei*

*et al., 2018*). Similarly, the VitroJet and Cryo-writer (pin-)print sub-nanoliter volumes directly onto a small area of the cryo-EM grid (*Ravelli et al., 2020*; *Arnold et al., 2017*). The cryoChip presented here further reduces the required sample volume by 1000-fold to only a few picoliters, approaching the practical minimum for liquid handling in cryo-EM structure determination. The cantilever-based sample application efficiently drives chip filling via capillary action, thus allowing essentially loss-less cryo-EM sample preparation. Together with the highly reproducible sample thickness we expect these properties to accelerate developments towards full automation of the cryo-EM sample preparation process. Notably, sample preparation using cryoChips does not require humidity control and is compatible with any standard plunger available in cryo-EM laboratories.

Using cryoChip supports, we demonstrated ab initio structure determination for three different test specimen resolved at resolutions suitable for mechanistic interpretation even from datasets with a comparatively small set of particles. The diversity in shape, size, and physico-chemical properties of the macromolecular complexes suggests that cryoChip sample preparation should be applicable for routine cryo-EM imaging of a wide variety of different macromolecular targets. Several aspects bear potential for future improvement. Despite our predictions from heat transfer simulations, we observed that between 10% and 40% of images contain non-vitreous ice. Several factors may contribute to this discrepancy. Firstly, it is possible that cryogen is displaced by the chip base upon entering the cryogen, preventing complete wetting of the chip; or that air is trapped in the depression on the chip base below the free-standing observation membrane, preventing contact with the coolant and impeding heat transfer from the nanochannels. Secondly, the observation membrane is located at the chip center which could affect probability of vitrification, consistent with the spatial distribution of isotherms reported for standard holey film grids during immersion cooling (*Kasas et al., 2003*). Parameters for existing plunging instruments have been optimised to maximise vitrification efficiency for holey film supports and it is conceivable that different plunging speed or depth, or jet-based cooling, may be more appropriate for cryoChips.

In standard cryo-EM sample preparation using holey support grids, the blotting process leaves a very thin layer of sample that can be imaged with high contrast and minimal secondary scattering from the particle-embedding ice layer. In cryoChips, the concave cross-sectional shape of the nanochannel produces a defined thickness gradient within which particles naturally partition into monolayers according to their molecular dimensions in the thinner parts of the channel, thus minimising background scattering from surrounding ice. The sample confinement in cryoChip nanochannels, however, comes with the penalty of additional incoherent scattering from the enclosing $SiN_x$ membranes, which results in reduced image contrast compared to ideal samples. By quantifying the additional scattering from the nanochannel membranes as an effective increase in ice thickness, we estimate the minimal scattering mass of particles that can still be successfully detected and aligned under these conditions to ~200 kDa (*Henderson, 1995*; *Zhang et al., 2020*). This should render cryoChips suitable for imaging a wide range of relevant protein targets for cryo-EM structure determination.

Apart from incoherent scattering contributions by the $SiN_x$ membrane, several other factors may currently limit resolution: Firstly, spectral properties of structural noise contributed by the $SiN_x$ membranes are different from those of vitreous ice, and secondly, there may be interference of signal from $SiN_x$ membranes and particles located at different focal heights within the nanochannel that may affect accuracy of CTF estimation (*Tichelaar et al., 2020*, *Figure 6—figure supplement 2*). It is possible that assumptions on image statistics in current 3D reconstruction algorithms are sub-optimal for particles from cryoChips and improvements could be gained with modified software.

Another general problem in cryo-EM is the degradation of image quality through electron radiation–induced particle movement (*McMullan et al., 2009*). Beam-induced movement is assumed to be the result of a complex combination of radiolysis, charge build-up around the irradiation area and mechanical stress release during irradiation (*Glaeser et al., 2011*; *Brilot et al., 2012*). Several of these factors are of relevance for cryoChips. First, nanochannels are closed volumes implying that molecular hydrogen formed as a product of electron-beam induced radiolysis (*Aronova et al., 2011*) cannot escape. Trapped gas may lead to pressure build-up inside the nanochannel and ultimately specimen movement. Second, $SiN_x$ is a poor conductor (resistivity $>10^{14}$ $\Omega cm$ at room temperature), suggesting that impaired charge dissipation could affect the observed specimen movement. While we found

overall specimen movement in cryoChips to be higher than for modified support films, the residual particle displacement is not limiting for ~2 Å reconstructions and hence in a range suitable for high-resolution map interpretation.

Tomographic reconstructions of sample-filled nanochannels showed that proteins preferentially partition to the solid-liquid interface of the nanochannel surface. This is consistent with observations using modified EM grids that contain surface substrates spanning the holey support film (*D'Imprima et al., 2019*; *Naydenova et al., 2019*; *Russo and Passmore, 2014*). While this may appear to raise issues similar to adsorption at the air-water interface, the nature of the interaction with these surfaces is fundamentally different. The air-water interface is hydrophobic and adsorption of proteins to the interface can drive partial or complete unfolding to minimise the free energy of interaction (*Glaeser, 2018*; *D'Imprima et al., 2019*; *Noble et al., 2018*). In contrast, similar to oxidised graphene, the polar SiN$_x$ surface is hydrophilic and adsorption of proteins is primarily mediated by the zeta potential of the interfacial electrical double layer (*Norde, 1986*). Consistently, for the three cases we investigated we do not observe that this interaction promotes denaturation or aberrant structural changes. It is entirely possible that the charge distribution between the molecular electrostatic potential and at the solid-liquid interface could promote preferential alignment and hence carry the risk of systematic bias in particle orientation. For the cases studied we did not find evidence for pronounced preferred orientation and could not identify strong correlation of enriched views with the molecular electrostatic potential.

We also observed that the strong interaction at the solid-liquid interface effectively concentrates the solute particles in the nanochannels, consistent with observations for other substrates providing solid-liquid interfaces such as hydrogenated graphene (*Russo and Passmore, 2014*). While this is principally beneficial for concentrating dilute protein solutions to particle densities suitable for imaging, it also affects design considerations for the filling channels. In early design iterations of the cryoChips that featured narrow filling and exit channels, we observed apparent depletion of the protein sample before it reached the observation nanochannels. This effect was further aggravated with samples containing high surface charge densities. We attribute this to the very large surface-to-volume ratio of nanochannels, which we suspect may bear the risk to deplete solute proteins through interactions with the SiN walls in the filling channel, thus gradually reducing the free protein concentration in solution as they travel towards the observation membrane. The increased diameter of the filling channel in the current cryoChip design substantially improved the efficacy for reaching suitable particle concentrations in the nanochannels of the observation membrane. With solid-liquid interfaces gaining importance also for existing cryo-EM supports (*D'Imprima et al., 2019*; *Naydenova et al., 2019*; *Naydenova et al., 2020*), further studies are needed to reveal the detailed dynamics and underlying mechanisms of particle adsorption to such interfaces, and to systematically investigate their effects on particle integrity and orientation.

Just as continuous developments have pushed the limit of achievable contrast and residual specimen movement for holey support films (*Naydenova et al., 2020*), we expect that similar progress can be attained for nanofluidic supports by further improvements in electron transparency, as well as electrical and thermal conductance of the enclosing materials. We anticipate that ongoing developments will eventually allow approaching resolutions and conditions realised for ideal samples (*Zivanov et al., 2018*).

We predict that the cryoChip is only the beginning of a range of future cryo-EM supports building on MEMS technology. Several immediate improvements can be achieved with minor modifications to our current design. Currently the observation membrane is 50 μm x 50 μm, representing less than 0.1% of the entire chip area. The imaging area can be readily extended by additional observation membranes, thus allowing fully automated collection of thousands of images with uniform ice thickness from a single chip. Parallel arrangements of observation membranes with separate filling channels will allow screening different samples on the same chip. Beyond this, the versatility of nanofabrication technology opens up possibilities to facilitate a range of new cryo-EM applications by bridging lab-on-a-chip approaches with structural biology. These include possibilities for on-chip mixing, parallelised high-throughput screening for structure-based drug discovery, time-resolved experiments, on-chip purification of complex protein mixtures and the combination of structure determination with biophysical measurements.

# Materials and methods

## Key resources table

| Reagent type (species) or resource | Designation | Source or reference | Identifiers | Additional information |
|---|---|---|---|---|
| Gene (*P. furiosus*) | Ferritin | Gift from Carsten Sachse | Uniprot ID:Q8U2T8 | |
| Gene (*T. acidophilium*) | T20S proteasome | Gift from Yifan Cheng | Addgene ID:110805 | |
| Software, algorithm | cryoSPARC | *Punjani et al., 2017* PMID:28165473 | RRID:SCR_016501 v3.1 | |
| Software, algorithm | UCSF ChimeraX | *Goddard et al., 2018* PMID:28710774 | RRID:SCR_015872 v1.2 | |
| Software, algorithm | Coot | *Emsley et al., 2010* PMID:20383002 | RRID:SCR_014222 v0.8.9.2 | |
| Software, algorithm | Phenix | *Adams et al., 2010* PMID:20124702 | RRID:SCR_014224 v1.13 | |
| Software, algorithm | MOLPROBITY | *Chen et al., 2010* PMID:20057044 | RRID:SCR_014226 v1.17.1 | |
| Software, algorithm | MotionCor2 | *Zheng et al., 2017* PMID:28250466 | RRID:SCR_016499 v1.0 | |
| Software, algorithm | IMOD | *Mastronarde and Held, 2017* PMID:2744392 | RRID:SCR_003297 v4.9.12 | |

## Chip fabrication

The cryoChip is fabricated using a wafer-scale surface micromachining process. The fluidic components, such as hollow cantilevers, thick-walled supply- and exit channels, and thin-walled nanochannels, are all formed by sacrificial polysilicon etching techniques. The fabrication process starts on a 4 inches, <100>-oriented silicon wafer of 380 µm thickness. The wafer is locally thinned from the backside to form silicon membranes with a thickness of 180 µm, in which later the cryoChip membrane will be formed. The backside processing is performed by anisotropic wet etching of silicon in a 25% potassium hydroxide (KOH) solution at 75°C. A 50-nm-thick stoichiometric silicon nitride ($Si_3N_4$) layer is used as the masking material for the KOH etch. After the KOH etching the masking layer is removed in concentrated hydrofluoric acid (HF 49%). On the locally thinned wafer, a 300-nm-thick silicon oxide layer is deposited in a low-pressure chemical vapour deposition (LPCVD) oven by pyrolysis of tetraethylorthosilicate (TEOS). The silicon oxide layer is patterned by buffered hydrofluoric acid (BHF). Silicon oxide is removed from an area where ultra-thin walled observation nanochannels will be formed. On the patterned substrate, an 11-nm silicon-rich nitride ($SiN_x$) layer is deposited by LPCVD. This low-stress $SiN_x$ layer serves as a bottom wall of the TEM observation windows, which will be created later. Next, a sacrificial layer of polysilicon with a thickness of 300 nm is deposited by LPCVD. This relatively thick polysilicon layer will be used to create the supply- and exit channels and a large fluidic channel inside the hollow cantilevers. The deposited polysilicon layer is patterned by tetramethylammonium hydroxide (TMAH). The layer is removed from a region where later the observation nanochannels will be formed. After the TMAH patterning, another sacrificial polysilicon layer with a thickness of 100 nm is deposited by LPCVD The thickness of the second sacrificial layer determines the height of the small fluidic channels and the height of the TEM observation windows. On the other hand, the aggregate thickness of the first and the second polysilicon layer, which is 400 nm in our design, determines the height of the larger fluidic channels. After the deposition, sacrificial polysilicon is patterned by TMAH etching to define an outline of the fluidic channels. A rather thin thermally grown silicon oxide layer (40 nm) is used as an etching mask. It should be noted that both large (400 nm) and small (100 nm) fluidic channels are etched at the same time. The underetching of the mask, due to an isotropic nature of the TMAH etch process, is used to reduce the width of the fluidic channels in which the TEM observation windows will be formed. After patterning of the fluidic channels, the second silicon-rich nitride layer is deposited by LPCVD. The 11-nm-thick low-stress $SiN_x$ layer will define a top wall of the TEM observation windows. Thereafter, a 300-nm-thick silicon oxide layer is deposited by TEOS pyrolysis. A multilayer of $SiN_x$ and silicon oxide will form the walls of the hollow cantilevers and some fluidic channels. The relatively large thickness of these walls provides mechanical stability of these components.

After patterning of the second TEOS layer in BHF the TEM observation windows are completely defined. Subsequently, the TEOS/SiN$_x$/SiN$_x$/TEOS multilayer is etched by reactive ion etching (RIE). In this processing step, the contour of the chip and the hollow cantilevers are patterned. After the RIE etching, an access to the bulk silicon is created. By using Deep Reactive Ion Etching (DRIE) the bulk silicon is etched from the front side to a depth of 150 µm. Finally, the wafer is placed in a 25% TMAH solution heated at 90°C. During the TMAH etching, sacrificial polysilicon is removed through the access holes and the fluidic channels are created. These temporary access holes are automatically sealed after final drying of the samples. At the same time, the bulk silicon is exposed to TMAH. A part of the bulk silicon underneath the TEM observation window is removed creating a suspended membrane with integrated thin-walled fluidic nanochannels. The bulk underneath the cantilevers is also removed and the cantilevers are released. During the TMAH etching the backside of the wafer remains unprotected. Due to the removal of silicon the cryoChip is further thinned down to its final thickness of 100 µm.

## Filling volume

The filling volume entering the chip through the cantilever was estimated from the chip geometry. The supply microchannel is about 1 mm long, 0.4 µm high, and 4 µm wide, which results in an estimated volume of ~2 pL. The five nanochannels for imaging are 50 µm long, ~750 nm wide, and 100 nm high, which (assuming a parabolic profile) yields an aggregate volume of 6.25 fL, negligible compared to the microchannel volume. Assuming the liquid fills the entire entry microchannel and exit microchannel, the total filling volume is ~4 pL.

## SEM imaging and FIB

SEM imaging of cryoChips was performed on a Helios G4 CX DualBeam FIB/SEM system (ThermoFisher Scientific) at ×35,000 magnification using the TLD detector. A gallium beam was used to mill a rectangular opening in the observation membrane to image the cross-section of the nanochannel. Grayscale SEM images were coloured in Adobe Photoshop according to the schematic representations in *Figure 1*.

## Thickness measurements of silicon nitride

For thickness determination of the ultrathin silicon nitride layer surrounding the nanochannels, we deposited a silicon-rich nitride layer on a dummy wafer using the same deposition parameters as used for cryoChip fabrication. The dummy wafer was annealed at 1100°C. Next, the wafer was etched for 1 min in HF 1% solution ('etch 1'). In our fabrication process, the HF 1% dip is used to remove a native oxide layer from silicon prior to the TMAH etchings. Then the dummy wafer was etched in TMAH 25% solution at 90°C ('etch 2'). The same etching time is used in our fabrication process to release the samples. The thickness of the SiN$_x$ layer after deposition and both etching steps was measured on 25 positions of the wafer by ellipsometry.

For thickness determination by energy-filtered TEM (EFTEM), we acquired images of a filled nanochannel at 8000x magnification (4.706 Å/pixel) on a JEOL JEM3200-FSC operated at 300 kV (K2 detector) with and without a 20 eV slit (centered at 0 eV) inserted below the omega filter. Images were recorded on a Gatan K2 direct electron detector. The resulting images were aligned and $I/I_0$ was computed by dividing the pixel values of the zero-loss image by those of the unfiltered image. The known total SiN thickness of 19 nm from ellipsometry measurements was used to calibrate the mean free path for inelastic scattering of silicon-rich nitride to 163 nm at 300 kV. We assumed a literature value of 320 nm (*Yesibolati et al., 2020*) for the mean free path of vitreous ice. The derived quantity 'water-equivalent thickness' ($WET$) is the thickness of water/vitreous ice that would lead to the same $I/I_0$, and is computed as follows:

$$WET = 320 \text{ nm} \cdot \ln \frac{I_0}{I} \tag{1}$$

The quantity 'ice thickness' ($IT$) is computed by removing the contribution of the silicon nitride from $WET$:

$$IT = WET - (163 \text{ nm} \cdot \ln \frac{1}{0.89}) \tag{2}$$

*WET* cross-sections of nanochannels of whole chips were computed from grid-maps of three chips at 2500x (72.64 Å/pixel) on a TVIPS XF416 detector with and without a 20 eV slit inserted. Observation windows in unfiltered and zero-loss images were aligned by cross-correlation, and profiles across each nanochannel computed as averages over five stripes per observation window. Additionally, the average over all profiles was computed for each chip.

## Protein expression and purification

*ApoFtn: Pyrococcus furiosus* apoferritin (Uniprot Q8U2T8, in pET-24d(+)) was expressed in *E. coli* BL21-DE3 cells using autoinduction (**Studier, 2005**). Cells were initially grown at 37°C for 4 hr after which temperature was reduced to 20°C. After 20 h, cells were harvested, resuspended in lysis buffer (50 mM Tris, 300 mM NaCl, 0.1% Triton X-100) and disrupted by sonication. The lysate was cleared by centrifugation at 17,000 g at 4°C for 45 min. The cleared supernatant was boiled for 20 min at 80°C and centrifuged for 45 min at 17,000 g at 4°C. The supernatant was dialysed into ion exchange buffer (20 mM NaCl, 20 mM HEPES, pH=6.8) overnight and loaded onto a 1 mL HiTrap Q XL column (Cytiva). Following a wash step with 25 column volumes (cv) ion exchange buffer, bound protein was eluted with a 20-500 mM NaCl gradient over 50 cv. Fractions containing target protein in high purity were selected by SDS-PAGE and pooled. Pooled fractions were concentrated and finally purified by size exclusion chromatography on a Superdex200 10/300 column (Cytiva) in 20 mM Tris, 50 mM NaCl, pH 7.5. Fractions of the elution peak were pooled and concentrated to a final concentration of 3.4 mg/mL.

*TMV:* TMV was purified from infected tobacco leaves as described (**Gooding and Hebert, 1967**) and concentrated to 35 mg/mL.

*T20S: Thermoplasma acidophilium* 20S proteasome (pRSF-T20S, Addgene plasmid #110805) was expressed and purified essentially as described previously (**Yu et al., 2010**). In brief, protein was expressed in *E. coli* BL21-DE3 cells using autoinduction medium following the same expression protocol as for ApoFtn. After 20 hr, cells were harvested and lysed by sonication in lysis buffer (50 mM NaH2PO4, 150 mM NaCl, pH 8.0), followed by addition of three volumes of boiling lysis buffer to precipitate non-thermostable proteins. The supernatant was further purified by immobilised metal affinity chromatography (IMAC) followed by cleavage of the His-tag by TEV protease at room temperature for 48 hr. Final polishing was done by size exclusion chromatography on a Superose 6 column (Cytiva) with SEC buffer (20 mM Tris, 50 mM NaCl, 0.1 mM EDTA, pH 8.0). Fractions of the elution peak were pooled and concentrated to 1.4 mg/mL.

## cryoChip filling and vitrification

A microvolume filling station for cryoChips was built on an optical breadboard (Thorlabs, USA). A 10 µL positive displacement microvolume glass syringe (Hamilton) was mounted on a xyz translation stage for precise position control of a sample droplet relative to the sample application cantilever of a cryoChip. Two Toolcraft USB-microscopes (Conrad Electronics GmbH, Germany) were used for a top and side view of the filling process with video acquisition control on a computer. The transparent front cap was removed from the microscopes to increase the working distance. Both microscopes were mounted on an xy translation stage to orient the field of view. All experiments were done under ambient conditions without temperature or humidity control. A cryoChip was gripped on its edge with tweezers and positioned on a metal bearing for rapid and reproducible positioning in the field of view. Approximately 0.2 µL of sample was aspirated with the microvolume syringe. The syringe tip was dried with a tissue to avoid wetting of the tip during droplet extrusion. The syringe was mounted on the loading station and a small droplet of ~10-40 nL was extruded from the syringe. Using the xyz stage, the droplet was brought into contact with the cantilever tip. Within seconds, the liquid fills the entry microchannel, the nanochannels and parts of the exit microchannel. The liquid in the syringe was typically sufficient to fill many chips by drying the tip again with a tissue and extruding a new droplet. We did not observe a change in hydrophilic character of the cryoChip or reduction in chip filling capacity over a period of several month. Approximately 2-3 s after immersing the cantilever tip into the sample droplet, the tweezer with the chip was removed from the loading station and transferred on a Leica EM GP2 vitrification robot set to 20°C. The plunger was set to blot for zero seconds and the chip was plunged immediately into liquid ethane kept at -180°C. The time between removing the tweezer and chip from the filling station and plunge freezing was ~10 s.

## Electron cryo-tomography

A cryoChip with ApoFtn at 3.4 mg/mL concentration was prepared as described above and stored in liquid nitrogen until imaging. Tilt series of cryoChips were acquired on a JEOL JEM3200-FSC with a bidirectional tilt scheme starting from 0° to -60° and subsequently from 2° to 60°, in increments of 2° with a total exposure of 94.6 e⁻/Å² , a pixel size of 4.706 Å and a defocus target of 6 μm. Micrographs acquired as dose-fractionated movies of 30 frames were motion-corrected in MotionCor2 v1.0 (*Zheng et al., 2017*) and dose-weighted according to their accumulated dose (*Grant and Grigorieff, 2015*). The tilt series was aligned using patch tracking and reconstructed using back-projection as implemented in Etomo from the IMOD package (*Mastronarde and Held, 2017*). Segmentation of silicon nitride membrane and particles was done in the tomoseg program of EMAN2 using a machine learning approach (*Chen et al., 2017*). Segmented tomograms were visualised in ChimeraX (*Goddard et al., 2018*). A subvolume of the segmented ApoFtn tomogram with 376 x 750 x 100 nm³ channel volume was segmented in ChimeraX to count individual ApoFtn particles for concentration determination using the 'segment map' command. Tomogram acquisition and reconstruction for cryoChips with TMV and T20S were done using the same protocol, using a pixel size of 4.706 Å for TMV and 3.668 Å for T20S.

## Single-particle imaging of ApoFtn

Three cryoChips with ApoFtn at 3.4 mg/mL were sent to EMBL Heidelberg for data acquisition. A grid map of the whole observation membrane was acquired for each chip. Acquisition points were set up to automatically acquire three images in a circle by beam-image shift in SerialEM (*Mastronarde, 2005*) using the 'multiple Record Setup' setting. Six acquisition points were placed in the nanochannel section of each observation window, which leads to 18 images per nanochannel section. Using py-EM (*Schorb et al., 2019*), reference maps were generated from the grid map to find back and align to the acquisition points with high precision. Coma-free alignment was performed on the collapsed silicon nitride membrane adjacent to the nanochannels. Movies of 90 frames were acquired during a 9 second exposure on a Krios (ThermoFisher Scientific) microscope with a Quantum-K2 detector. The total exposure of the image was 63 e⁻/Å², the pixel size 0.8127 Å/pix and the dose rate ~2 e⁻/pixel/s. The defocus range was set to -1 to -2 μm. In total 945 movies were acquired from three cryoChips. A reference dataset of 276 movies were acquired for ApoFtn on QF1.2/1.3 holey carbon grids using the same data acquisition settings.

## Single-particle imaging of T20S and TMV

Datasets comprising 121 movies of the T20S proteasome at 1.4 mg/mL and 82 movies of TMV at 1.1 mg/mL were acquired on a JEOL JEM 3200-FSC microscope operated at 300 kV. For T20S, we used cryoChips with a slightly different design featuring orthogonal nanochannels arranged in a cloverlike pattern instead of straight nanochannels (*Figure 6—figure supplement 1*). Both TMV and T20S single-particle datasets were acquired on a Quantum-K2 detector at a magnification of 30,000x (1.288 Å/pixel) and an exposure time of 10.95 seconds with a total exposure of 59.68 e⁻/Å² over 73 frames. A reference dataset of 879 movies were acquired for T20 on QF1.2/1.3 holey carbon grids using the same data acquisition settings.

## Single-particle image processing

From three cryoChips filled with ApoFtn, 945 movies were processed in cryoSPARC 3.1 (*Punjani et al., 2017*; *Figure 4—figure supplement 4*). Patch-motion correction and patched CTF estimation was followed by manual curation of micrographs to remove images with visible ice reflections in the power spectra, retaining 815 micrographs. Particles were picked with the blob-picker and subjected to 2D classification to obtain a picking reference. The reference was low-pass filtered to 20 Å and was used to pick 318,058 ApoFtn particles. After 2D classification, selection and local-motion correction in cryoSPARC, 97,431 particles were retained. Three more rounds of 2D classification were used to select for classes with visible secondary structure features, resulting in 34,102 particles. Retained particles were used for initial model generation by cryoSPARC's stochastic gradient descent method while imposing O symmetry. The resulting reference volume was low-pass filtered to 15 Å and used in homogeneous refinement, leading to a 3D reconstruction at 3.7 Å resolution. Per-particle defocus refinement and beam-tilt refinement improved the resolution to 3.0 Å. Heterogenous refinement with three classes

was used to further prune particles and lead to a 3.0 Å resolution reconstruction from 21,238 particles. In all cases, O symmetry was imposed during reconstruction. 276 movies of a reference dataset of ApoFtn on QF1.2/1.3 holey carbon grids were processed in the same way and led to a reconstruction at 2.4 Å resolution.

Eighty-two movies from two cryoChips filled with TMV were imported in cryoSPARC 3.1 (*Punjani et al., 2017*), followed by patch-motion correction and patched CTF estimation (*Figure 4—figure supplement 5*). Of 82 movies with vitreous ice, 43 were retained for further processing. Manual picking and 2D classification was used to obtain picking references. Filament tracing was used to trace and segment virus particles with a step size of 20 asymmetric units (1.41 Å each), resulting in 31,623 segments. Two rounds of 2D classification were used to prune segments. Local motion correction and re-extraction yielded 14,238 TMV segments. Helical refinement was started from a cylindrical initial model and an initial symmetry of 22.04° twist and 1.41 Å rise, while refining the symmetry. This yielded a 3D reconstruction at 4.3 Å reconstruction. Two rounds of per-particle defocus and global CTF (tilt & trefoil) refinement improved the reconstruction to 3.7 Å. The final refined symmetry converged to 22.036° twist and 1.415 Å helical rise.

A total of 121 movies of T20S proteasome were processed in cryoSPARC 3.1 (*Punjani et al., 2017*, *Figure 4—figure supplement 5*). Patch-motion correction and patched CTF estimation were performed, followed by manual curation of micrographs to remove movies with visible reflections from crystalline ice in their power spectra. 91 movies were retained for further processing. Blob-picking and 2D classification was used to generate references for automated picking. Five reference classes were low-pass filtered to 20 Å for template picking, resulting in 47,484 particles. After two rounds of 2D classification and selection, 12,351 particles were retained and used for ab-initio 3D reconstruction imposing D7 symmetry. The initial model thus obtained was low-pass filtered to 30 Å and used for homogenous refinement, yielding a reconstruction at 7.9 Å resolution. After heterogeneous refinement with three classes, particles in the class with highest resolution (5,750 particles) were selected and subjected to another round of homogenous refinement yielding the final reconstruction at 6.0 Å resolution. Per-particle defocus and global CTF refinement (tilt & trefoil) further improved the resolution to 5.4 Å. All reconstructions had D7 symmetry imposed. 879 movies of a reference dataset of T20S on QF1.2/1.3 holey carbon grids were processed in the same way and led to a final reconstruction at 3.2 Å resolution. For all final reconstructions, five independent runs of ResLog (*Stagg et al., 2014*) analysis were performed and the results averaged for ResLog curves. Local resolution was estimated in cryoSPARC using a windowed FSC method (*Cardone et al., 2013*) and visualised in ChimeraX (*Goddard et al., 2018*). Map sharpening was done in cryoSPARC by applying the overall B-factor estimated from Guinier plots.

## Atomic model building

Atomic models for TMV (PDB ID 4udv) and T20S (PDB ID 3j9i) were rigid body-fitted into the density maps and not further refined. The atomic model from an available crystal structure of ApoFtn (PDB ID 2x17) (*Kasyutich et al., 2010*) was rigid body-fitted into the cryo-EM density, and real-space refinement in Phenix 1.13 (*Adams et al., 2010*) was iterated with manual model building in Coot 0.8.9.2 (*Emsley et al., 2010*). Model validation statistics were retrieved from MOLPROBITY (*Chen et al., 2017*) as implemented in Phenix and overfitting was probed by half-map refinement.

## Beam-induced motion trajectory analysis

Four datasets (10200 [*Zivanov et al., 2018*], 10272 [*Naydenova et al., 2019*], 10306 [*Weis et al., 2019*], 10389 [*Righetto et al., 2020*]) from the Electron Microscopy Public Image Archive (EMPIAR [*Iudin et al., 2016*]) were subjected to full-frame motion correction in cryoSPARC (*Punjani et al., 2017*) to calculate motion trajectories and were compared with equivalently derived trajectories from the cryoChip ApoFtn dataset.

## Kymograph generation

Movies of the filling process for each cryoChip were acquired with a Toolcraft USB-microscope (Conrad Electronics GmbH, Germany) at 10 frames per second and a pixel size of 1.6 µm. A 10 pixel-wide section along the supply or exit microchannels was averaged and plotted over time. For each time point, coordinates of the liquid front were manually picked and individually plotted for each chip.

A square root function of the form $x(t) = \sqrt{a(t-b)} + c$ was fitted to each filling trajectory using the curve_fit function from the scipy package (*Virtanen et al., 2020*). For error estimation of the fitting parameters, a bootstrapping approach was used, performing the fit 100 times with a random set of $n$ out of $n$ coordinates with replacement. The parameter and its uncertainty was estimated as mean and standard deviation over the 100 fits. For the linearised plots of $x^2$ versus $t$, the shift parameter $c$ was subtracted from the curve.

### Convective heat transfer simulations

Simulations were performed in COMSOL Multiphysics 5.4 using a simplified 2D geometry of cryo-Chips represented by a 2 mm x 100 µm silicon base and 50 µm x 20 nm SiN observation membrane. The initial temperature of the chip was set to room temperature (293.15 K) and the ethane bath was at 90 K. Heat flux was modelled by a external forced convection model with 1.5 m/s fluid velocity and a plate length of 2 mm. This does not take into account geometry of the plunging process or fluid dynamics and uses an average heat flux for every surface of the chip. Temperature development over time was evaluated over a profile along the observation membrane and plotted in different time windows. Material properties were obtained from the COMSOL material library unless stated otherwise. The specific heat capacity $c_p$ for silicon nitride was assumed as 700 J/(kg K) and its density as 3100 $\mathrm{kg/m^3}$. For thermal conductivity of SiN, we averaged values from *Ftouni et al., 2015* for thin membranes over the relevant temperature range to obtain 2.3 W/(m K). For silicon, heat capacity was assumed as 700 J/(kg K), thermal conductivity as 130 W/(m K) and density as 2329 $\mathrm{kg/m^3}$.

## Acknowledgements

We thank Yifan Cheng for the expression plasmid of T20S proteasome and Carsten Sachse for the expression plasmid of ApoFtn. We acknowledge the EMBL cryo-EM imaging platform for access to their facilities, the staff from the Kavli Nanolab Delft for cleanroom support and the Department of Bionanoscience for generously supporting our cryo-EM facility. We thank Cecilia de Agrela Pinto and Wim Hagen for critically reading the manuscript and helpful suggestions; and Jacob Hoogenboom for insightful discussion. This work was supported by the European Research Council (ERC-StG-852880 to AJ), the Dutch Research Council (NWO.STU.018–2.007 to AJ) and the Kavli Institute of Nanoscience Delft.

## Additional information

### Competing interests

Edin Sarajlic, Roeland Huijink: ES and RH are inventors on a provisional patent application by SmartTip B.V. (NL2026627) for the use of microcantilevers with nanofluidic sample cells. The other authors declare that no competing interests exist.

### Funding

| Funder | Grant reference number | Author |
|---|---|---|
| Nederlandse Organisatie voor Wetenschappelijk Onderzoek | NWO.STU.018-2.007 | Arjen J Jakobi |
| European Research Council | ERC-StG-852880 | Arjen J Jakobi |

The funders had no role in study design, data collection and interpretation, or the decision to submit the work for publication.

### Author contributions

Stefan T Huber, Conceptualization, Formal analysis, Investigation, Methodology, Validation, Visualization, Writing – original draft, Writing – review and editing; Edin Sarajlic, Roeland Huijink, Conceptualization, Investigation, Methodology, Writing – review and editing; Felix Weis, Investigation, Writing

– review and editing; Wiel H Evers, Investigation, Writing – review and editing, Conceptualization, Formal analysis, Methodology; Arjen J Jakobi, Conceptualization, Formal analysis, Investigation, Methodology, Writing – review and editing, Funding acquisition, Project administration, Resources, Supervision, Validation, Visualization, Writing – original draft

#### Author ORCIDs
Stefan T Huber http://orcid.org/0000-0003-3721-5104
Wiel H Evers http://orcid.org/0000-0002-3413-5128
Arjen J Jakobi http://orcid.org/0000-0002-7761-2027

#### Decision letter and Author response
Decision letter https://doi.org/10.7554/eLife.72629.sa1
Author response https://doi.org/10.7554/eLife.72629.sa2

---

## Additional files

#### Supplementary files
• Transparent reporting form
• Supplementary file 1. Data collection and refinement statistics for collected datasets.
• Supplementary file 2. Data collection and refinement statistics for reference datasets on holey carbon grids.
• Supplementary file 3. Model refinement statistics for ApoFtn atomic model.
• Supplementary file 4. Data and 3D reconstruction parameters of reference datasets.

#### Data availability
Atomic coordinates have been deposited in the Protein Data Bank (PDB) under accession code 7ohf. Cryo-EM densities have been deposited in the Electron Microscopy Data Bank (EMDB) under accession codes EMD-12901, EMD-12903 and EMD-12915. Tomographic reconstructions have been deposited in the EMDB under accession codes EMD-12902, EMD-12914 and EMD-12917. Raw micrographs for all cryo-EM data have been deposited in the Electron Microscopy Public Image Archive (EMPIAR) under accession code 10708. Numerical data supporting graphs in Figures 1, 4, 5, Figure 2-Figure supplement 1, Figure 4-Figure supplement 2 and Figure 5-Figure supplement 1 are provided in Source Data files.

The following dataset was generated:

| Author(s) | Year | Dataset title | Dataset URL | Database and Identifier |
|---|---|---|---|---|
| Huber ST, Sarajlic E, Huijink R, Weis F, Evers WH, Jakobi AJ | 2021 | Apoferritin, TMV and T20S proteasome in nanofluidic channels | https://www.ebi.ac.uk/pdbe/emdb/empiar/entry/10708/ | Electron Microscopy Public Image Archive, EMPIAR-10708 |
| Huber ST, Sarajlic E, Huijink R, Evers WH, Jakobi AJ | 2021 | Cryo-EM structure of pyrococcus furiosus apoferritin in nanofluidic channels | https://www.rcsb.org/structure/7ohf | RCSB Protein Data Bank, 7ohf |
| Huber ST, Sarajlic E, Huijink R, Evers WH, Jakobi AJ | 2021 | Nanochannel filled with pyrococcus furiosus apoferritin | https://www.ebi.ac.uk/emdb/EMD-12902 | EMDB, EMD-12902 |
| Huber ST, Sarajlic E, Huijink R, Evers WH, Jakobi AJ | 2021 | Nanochannel filled with tobacco mosaic virus | https://www.ebi.ac.uk/emdb/EMD-12914 | EMDB, EMD-12914 |
| Huber ST, Sarajlic E, Huijink R, Evers WH, Jakobi AJ | 2021 | Thermoplasma acidophilium 20S proteasome | https://www.ebi.ac.uk/emdb/EMD-12917 | EMDB, EMD-12917 |

The following previously published datasets were used:

| Author(s) | Year | Dataset title | Dataset URL | Database and Identifier |
|---|---|---|---|---|
| Zivanov J, Nakane T, Forsberg B, Kimanius D, Hagen WJH, Lindahl E, Scheres SHW | 2018 | New tools for automated high-resolution cryo-EM structure determination in RELION-3 | https://www.ebi.ac.uk/pdbe/emdb/empiar/entry/10200/ | Electron Microscopy Public Image Archive, EMPIAR-10200 |
| Naydenova K, Peet MJ, Russo CJ | 2019 | Multifunctional graphene supports for electron cryomicroscopy | https://www.ebi.ac.uk/pdbe/emdb/empiar/entry/10272/ | Electron Microscopy Public Image Archive, EMPIAR-10272 |
| Weis F, Beckers M, von der Hocht I, Sachse C | 2019 | Elucidation of the viral disassembly switch of tobacco mosaic virus | https://www.ebi.ac.uk/pdbe/emdb/empiar/entry/10306/ | Electron Microscopy Public Image Archive, EMPIAR-10306 |
| Righetto RD, Anton L, Adaixo R, Jakob RP, Zivanov J, Mahi MA, Ringler P, Schwede T, Maier T, Stahlberg H | 2020 | High resolution cryo-EM structure of urease from the pathogen Yersinia enterocolitica | https://www.ebi.ac.uk/pdbe/emdb/empiar/entry/10389/ | Electron Microscopy Public Image Archive, EMPIAR-10389 |

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
