## [Editor Report]

In their pioneering work, the authors describe a prototype of a new microfluidic device that addresses several of the common sample preparation problems in single-particle cryo-EM. By providing small cavities made of thin silicon nitride membranes, picoliters of sample can be frozen with defined ice thickness while avoiding sample damage at the air-water interface. The new approach requires further development to become mainstream but has the potential to become a powerful alternative to the conventional workflow of single-particle cryo-EM, enabling full automation and making sample preparation highly reproducible.

---

## [Decision Letter]

**Decision letter after peer review:**

Thank you for submitting your article "Nanofluidic chips for cryo-EM structure determination from picoliter sample volumes" for consideration by *eLife*. Your article has been reviewed by 3 peer reviewers, including Nikolaus Grigorieff as Reviewing Editor and Reviewer #1, and the evaluation has been overseen by Olga Boudker as the Senior Editor. The following individuals involved in review of your submission have agreed to reveal their identity: Matthias Wolf (Reviewer #2); Henning Stahlberg (Reviewer #3).

*Reviewer #1 (Recommendations for the authors):*

1) The authors introduce a "combined water-equivalent thickness", which is a nice way of looking at the scattering contribution of the membranes to the images. On page 6, the authors write "This effect is equivalent to an apparent increase in ice thickness", which is not entirely true since the molecular motion of water seen under the electron beam will be absent in SiN, leading to very different spectral properties of the background (the authors acknowledge this on page 9). Would it be possible to characterize the background generated by the thin SiN membranes more quantitatively by looking at Fourier spectra?

2) Is there a reason for not choosing a membrane protein as one of the application examples? Perhaps the authors were worried about the detergent affecting the loading of the sample. If so, this should be discussed.

3) Page 1: It might be better to delete "to about 30 Å" at the end of the paragraph since it is difficult to isolate the cause of this limited resolution to the membranes.

4) Figure 4: Could the authors show an apoferritin class average along the 3-fold axis? If they do not see one, they should point this out as this would indicate a "preferred orientation" problem.

5) Page 6: The authors should explain the difference between the B-factor and the Debye-Waller factor.

6) Page 6: The authors state that the observed B-factors are in the range of what is observed in other studies. However, in studies of apoferritin, TMV and proteasomes, the B-factors are typically much smaller. Presumably, the B-factors depend primarily on the blurring (beam-induced motion) of the sample, and not so much on the background added by the SiN membranes. The authors should discuss why their B-factors are higher than those observed in other studies of these samples, and also why the proteasome reconstruction is so much worse than the other two reconstructions.

7) Figure 6: As the authors point out, the tomogram suggests that particles adhere to the solid-liquid interface. Does this lead to a depletion of particles in cavities that are located further away from the filling cantilever? If the authors have data to answer this question, this might be useful to show.

8) Page 9: The authors state that they do not observe denaturation of their samples at the solid-liquid interface. They should describe what they expect a denatured sample to look like, and how they would detect this.

*Reviewer #2 (Recommendations for the authors):*

Congratulations – excellent work!

If you have any data about how long your chips can retain their hydrophilic character during storage before use, this may be a valuable addition.

*Reviewer #3 (Recommendations for the authors):*

Concerning the ice in the samples, the authors should not quantify how many images had vitreous ice, because most images had. Instead, the authors should quantify how many images shows ice crystals, in addition to the vitreous areas. Is that cubic or hexagonal ice? Does it melt rapidly during imaging (as surface contamination sometimes does), or is it stable under the bean and therefore in the bulk sample?

---

## [Author Response]

Reviewer #1 (Recommendations for the authors):1) The authors introduce a "combined water-equivalent thickness", which is a nice way of looking at the scattering contribution of the membranes to the images. On page 6, the authors write "This effect is equivalent to an apparent increase in ice thickness", which is not entirely true since the molecular motion of water seen under the electron beam will be absent in SiN, leading to very different spectral properties of the background (the authors acknowledge this on page 9). Would it be possible to characterize the background generated by the thin SiN membranes more quantitatively by looking at Fourier spectra?

We thank the reviewer for this thoughtful comment. We followed the suggestion of the reviewer and analysed the spectral properties of SiN. In Figure 6 —figure supplement 2 we compare Fourier spectra of the 19 nm SiN membranes next to the nanochannels, containing only SiN and no ice or protein and Fourier spectra from ice-filled nanochannels with 9.5 nm SiN top and bottom membranes. Unlike amorphous carbon for which scattering power monotonically decreases with increasing spatial frequency, for SiN we find a broad scattering peak centred around 1/3.9 Å^-1^, indicating correlations in the local structure of the amorphous SiN. These spectra also illustrate the interference of Thon ring modulations from the indepentently scattering top and bottom membranes, leading to nodes in the overall CTF that may affect accurate CTF estimation.

2) Is there a reason for not choosing a membrane protein as one of the application examples? Perhaps the authors were worried about the detergent affecting the loading of the sample. If so, this should be discussed.

We agree with the reviewer that it will be highly interesting to test the possibilities and limitations of cryoChip supports for membrane proteins. We did not explore these in the current report as we had no such sample available and experimental opportunities were limited due to restrictions related to the COVID-19 pandemic. Exploring the utility of cryoChip as sample support for a broad range of biological specimen will be an important direction in future studies.

3) Page 1: It might be better to delete "to about 30 Å" at the end of the paragraph since it is difficult to isolate the cause of this limited resolution to the membranes.

We agree and have removed this part of the sentence.

4) Figure 4: Could the authors show an apoferritin class average along the 3-fold axis? If they do not see one, they should point this out as this would indicate a "preferred orientation" problem.

We have now included all 2D class averages of ApoFtn in Figure 4 —figure supplement 4. We do not see a class average precisely aligned along the three-fold axis, but we do see class averages representative of orientations different from that aligned along the 4-fold axis. The orientation distribution plot in Figure 4 —figure supplement 4 indicates, as stated in the manuscript, a slight overrepresentation of views along the 4-fold axis, which is similarly frequently observed for this specimen on conventional sample supports.

5) Page 6: The authors should explain the difference between the B-factor and the Debye-Waller factor.

We realise that the original sentence on p.6 was confusing and have rephrased it as follows:

“The slope of this function is related to a B-factor that describes the width of a Gaussian dampening function in reciprocal space. This B-factor is used to model the loss of high-resolution information in particle images owing to a combined effect of performance-reducing factors related to the sample, data collection, and data processing (Rosenthal and Henderson, 2003).”

6) Page 6: The authors state that the observed B-factors are in the range of what is observed in other studies. However, in studies of apoferritin, TMV and proteasomes, the B-factors are typically much smaller. Presumably, the B-factors depend primarily on the blurring (beam-induced motion) of the sample, and not so much on the background added by the SiN membranes. The authors should discuss why their B-factors are higher than those observed in other studies of these samples, and also why the proteasome reconstruction is so much worse than the other two reconstructions.

To investigate this point further, we also acquired reference datasets for ApoFtn and T20S on conventional holey carbon support films on the same microscope and using the same acquisiton settings. The results of these experiments are summarised in a new Figure 4 —figure supplement 7/Supplementary Table 2. As correctly assumed by the reviewer, we find the B-factors for cryoChip datasets to be higher than for the refernce datasets from holey carbon supports. A comparative analysis of the motion trajectories for both EM supports underpins the assumption that this difference is predominantly related to an increase in overall beam-induced motion for cryoChips (Figure 4 —figure supplement 7b,c). One additional difference we observe is a consistent upward shift of the B-factor plots for the holey carbon datasets. The slope of the Bfactor plot represents the information decay as a function of spatial frequency, while the offset of the curve is related to the overall signal-to-noise ratio in the data. The upward shift of the B-factor plots for data from holey carbon films is thus consistent with a increase in image contrast for these data compared to data acquired on cryoChips. We added the following paragraph to the Results section:

“For comparison, we also acquired datasets for ApoFtn and T20S on conventional holey carbon supports on the same microscope and with the same acquisition parameters, yielding reconstructions at an overall resolution of 2.4 and 3.2 Å (Figure 4 —figure supplement 7, Supplementary Table 2). We computed B-factor plots for all datasets resulting in B-factors of 217 Å2/178 Å2 for ApoFtn and 490 Å2/328 Å2 for T20S determined for cryoChip and holey carbon datasets, respectively. We also note a consistent upward shift of the offset for B-factor plots from holey carbon compared to those from cryoChips, consistent with the differences in overall signal-to-noise ratio caused by additional scattering from the SiNx membranes.”

7) Figure 6: As the authors point out, the tomogram suggests that particles adhere to the solid-liquid interface. Does this lead to a depletion of particles in cavities that are located further away from the filling cantilever? If the authors have data to answer this question, this might be useful to show.

We do not observe obvious differences in concentration throughout the nanochannels. In earlier chip iterations not reported in the manuscript, which featured shorter supply and exit channels, we observed a gradient of particle depletion from the start towards the end of the observation channels, suggesting adherence of particles to the solid-liquid interface (as conjectured by the reviewer, and also mentioned in the original discussion (p.10)). In the current version of the cryoChip, the liquid passes through a 1 mm long entry channel with a larger cross-section, is then drawn through the observation cavities, and exits into a 1 mm of exit channel with large cross-section. We assume that drawing a large enough volume through the cavities will lead to saturation of the solid-liquid interfaces in the cavity walls.

8) Page 9: The authors state that they do not observe denaturation of their samples at the solid-liquid interface. They should describe what they expect a denatured sample to look like, and how they would detect this.

We agree with the reviewer that we do not have definite proof for the absence of sample denaturation. We base our statement on indirect evidence from the absence of 2D classes representing partial complexes (e.g. proteasomes with less than 14 units, or broken apoferritin shells). The fact that we achieve high-resolution 3D reconstructions from a significant fraction of overall particles further suggests that denaturation cannot be a very prominent phenomenon at least for the samples studied here.

Reviewer #2 (Recommendations for the authors):Congratulations – excellent work!If you have any data about how long your chips can retain their hydrophilic character during storage before use, this may be a valuable addition.

We have not systematically tested this, but have been reusing cryoChips over a period of >6 months after storage in the lab without further precautions and did not find evidence for reduced filling capacity over this period. We have added the following sentence to the Methods section:

“We did not observe a change in hydrophilic character of the cryoChip, or reduction in chip filling capacity over a period of several months.”

Reviewer #3 (Recommendations for the authors):Concerning the ice in the samples, the authors should not quantify how many images had vitreous ice, because most images had. Instead, the authors should quantify how many images shows ice crystals, in addition to the vitreous areas. Is that cubic or hexagonal ice?

Our way of categorising images into non-vitreous/vitreous was chosen because it is a typical part in our processing workflow for any cryo-EM dataset, where we usually exclude images with strong ice reflections in their Fourier spectra. This might lead to exclusion of too many images, as some might have crystalline surface ice contamination, or only partial crystallinity.

We analysed some of the micrographs categorised as containing crystalline ice (summarised in Figure 4 —figure supplement 3) and show that they are almost entirely crystalline. For a subset of images, we tried to discriminate whether the ice is cubic or hexagonal. Pure hexagonal ice has 7 diffraction rings between 4 and 1.6 Å resolution, whereas cubic ice has three diffraction rings, the locations of which conincide with those of the (002), (110) and (112) hexagonal ice rings. We used the strong (100) reflection at 3.90 Å to uniquely identify hexagonal ice. All images had a strong (100) reflection corresponding to hexagonal ice; due to the overlap of cubic ice reflections with the hexagonal lattice and the weak intensity of (102) and (103) reflections in Fourier spectra computed from images, we cannot confidently say whether the images do also contain cubic ice. For a global analysis, we computed a maximum intensity projection of all Fourier spectra from the entire ApoFtn dataset to collect all ice reflections combined in a single Fourier spectrum. A rotational average of this spectrum displays a strong (100) reflection at 3.90 Å. In view of these data, we suggest that most likely explanation for non-vitreous areas is that the cooling rate at some parts of the observation membrane was too slow to enable glass transition, and that the predominant ice type is hexagonal. As discussed, we cannot exclude that non-vitreous areas contain a mix of hexagonal and cubic ice, but suggest that this is less likely as it would require heating of the specimen above the glass transition temperature during handling, or during exposure.

We added the following sentences to the Results section on p.7:

“To investigate whether non-vitreous images are a result of inefficient vitrification or phase transition to cubic ice during exposure, we analysed a subset of images classified as icy. From a maximum intensity projection of all power spectra of all images, we find evidence for hexagonal ice indicating incomplete vitrification (Figure 4 —figure supplement 3).”

Does it melt rapidly during imaging (as surface contamination sometimes does), or is it stable under the bean and therefore in the bulk sample?

As suggested by the reviewer, we examined Fourier spectra of a subset of videos from the ApoFtn dataset as a function of exposure and did not observe significant intensity changes for ice reflections (Figure 4 —figure supplement 3b). This favours a scenario in which the ice reflections originate from crystalline ice in the channels and not from surface ice.